# An Improved Wake-Up Receiver Based on the Optimization of Low-Frequency Pattern Matchers

**DOI:** 10.3390/s23198188

**Published:** 2023-09-30

**Authors:** Robert Fromm, Olfa Kanoun, Faouzi Derbel

**Affiliations:** 1Smart Diagnostic and Online Monitoring, Leipzig University of Applied Sciences, Wächterstraße 13, 04107 Leipzig, Germany; 2Measurement and Sensor Technology, Chemnitz University of Technology, Reichenhainer Straße 70, 09126 Chemnitz, Germany

**Keywords:** wireless sensor network, radio frequency, ultra-low power, Schottky diode, packet error rate, on-demand communication

## Abstract

Wake-up receivers are gaining importance in power-aware wireless sensor networks, as they significantly reduce power consumption during RF reception, enabling asynchronous communication with low latency. However, the performance of wake-up receivers still lags behind that of off-the-shelf RF transceivers. There is a growing demand for higher sensitivity, enhanced reliability, and lower latency while maintaining the lowest power consumption. In this article, our goal is to advance the performance of wake-up receivers based on off-the-shelf components and low-frequency pattern matchers. Through a systematic investigation, we proposed multiple improvements aimed at enhancing wake-up receiver performance and reliability. We introduced an improved passive envelope detector and realized a wake-up receiver for the 868 MHz band, which achieves a power consumption of 5.71 μW and latency of 9.02 ms. Our proposed wake-up receiver is capable of detecting signals down to an average power level of −61.6 dBm. These achievements represent significant advancements compared to the existing state of research on wake-up receivers based on low-frequency pattern matchers. Recent articles have not been able to attain such improved values in signal detection, power consumption, and latency.

## 1. Introduction

Wireless sensor networks (WSNs) are steadily gaining importance in both research and industry. They have become crucial for sensing and collecting environmental data. Using batteries to power sensor nodes is often necessary because recharging or replacing batteries can be impractical and would significantly increase maintenance costs. Furthermore, latency, transmission range, and the minimum detectable signal (MDS) are also key parameters in the hardware design of a sensor node [1].

Many applications require an autonomous WSN that maintains a continuous low-latency wireless communication. Even modern radio frequency (RF) transceivers consume 10–100 mW to maintain such a wireless connection. To power a sensor node with a battery for an extended period, it is necessary to significantly reduce the intervals for receiving and transmitting data. Typically, a duty-cycling approach is employed, but this approach has the drawback of increasing the latency and response times of the entire WSN. Our proposed solution is to integrate a wake-up receiver (WuRx). The WuRx’s power consumption should be in the range of 10 μW to enable continuous reception, even with limited battery size [2].

In Figure 1, the integration of a WuRx into a sensor node is shown. A sensor node typically comprises components such as an antenna, wireless transceiver, sensors, a power source, and a microcontroller. The WuRx is a specialized type of RF receiver designed to detect specific wake-up packets (WuPts). Its addition introduces a second path for RF reception within the sensor node. Either the wireless transceiver receives data packets or the WuRx captures WuPts. To switch between these two RF paths, multiple antennas can be utilized, or an RF switch can be introduced. The microcontroller is responsible for configuring the WuRx. During standby mode, the WuRx is the only active component in the circuit. If a WuPt is detected, the WuRx sends an interrupt signal to the microcontroller, prompting it to wake up [2].

WuRxs can be categorized into two groups: ASIC-based (application-specific integrated circuit) WuRxs and WuRxs implemented using commercial off-the-shelf (COTS) components [3]. This article will concentrate on the COTS-based implementations of WuRxs. This choice is based on several factors, including the improved repeatability of results, simpler and more cost-effective implementation, and the feasibility of integrating a WuRx into a commercial product.

When using COTS components, various approaches exist for constructing the architecture of the WuRx. However, a shared feature among most COTS architectures is the utilization of a passive envelope detector constructed with Schottky diodes [1,4]. Envelope detectors are only capable of detecting amplitude-modulated signals, which is why most implementations use on-off keying (OOK) modulation as a simplified form of amplitude modulation. The amplification and digitization of the envelope detector signal can significantly vary across these architectures.

In the implementations described in [5,6], comparator circuits are employed to digitize the low frequency (LF) signal. However, the specific method used for pattern matching and the process for generating a dependable wake-up signal are not addressed in these articles. The implementation described in [6] achieves a MDS of −70 dBm but requires additional components to amplify the LF signal.

References [7,8,9] employ low-noise amplifiers to boost the RF signal and improve the MDS. Nevertheless, these designs require several supplementary components, leading to larger board space requirements and elevated material costs. Furthermore, these architectures utilize a duty-cycling approach, which results in increased latency.

This article focuses on the utilization of components known as low-frequency pattern matchers (LFPMs). LFPMs are specialized components capable of detecting specific patterns within LF signals. WuRxs based on LFPMs are known for having a reliable communication protocol with a low probability of false wake-ups. Thanks to the integrated LFPM, these WuRxs can be implemented using a minimal number of components [2]. Our proposed WuRx design, which incorporates only eight discrete surface-mounted devices, results in a compact board size and lower material costs. Figure 2 shows the building blocks of such a WuRx, which utilizes a passive envelope detector and an LFPM.

The WuRx’s input signal is characterized by low amplitude, high noise figure, and various interferences [4]. An RF band-pass filter can be introduced to pass only signals of the desired frequency band. The envelope detector performs the signal detection and conversion to an LF signal. An impedance-matching circuit is needed to avoid power losses through signal reflection from the envelope detector. An LF amplifier circuit can be introduced to boost the rectified signal. The output signal is an LF signal, which is detectable by the following LFPM. The LFPMs are designed for detecting special LF signal and matching a pattern of the demodulated signal. An address pattern can be configured and the LFPM is capable of generating a digital output signal. On a successful pattern match, the wake-up signal is generated [2,10].

In all the proposed WuRx solutions, there exists an inevitable trade-off involving three key parameters: MDS, power consumption, and latency. Achieving an improved MDS in a WuRx allows for an extended transmission range for the WuPt. This increased range can also be achieved by raising the transmission power of the wake-up transmitter (WuTx). However, this approach is often constrained either by legal regulations or the increased power consumption in the WuTx. To maximize the battery life of the sensor node, it is crucial to minimize the power consumption of the WuRx. Likewise, it is important to keep the WuRx latency as short as possible to gain an advantage over non-WuRx communication. Some specific applications, such as localization, require a latency of 10 ms or less [11,12]. For an always-on WuRx, the latency approximately equals the duration of the WuPt. Additionally, a longer WuPt duration results in higher power consumption in the WuTx, which becomes particularly significant when the WuTx is also battery-powered. This is typically the case in multi-hop networks.

Particularly when using the integrated circuit AS3933 as the LFPM, there are multiple configuration settings available [13]. However, in recent publications (see Section 2.1), a comprehensive exploration of the effects of these settings on reliability and the performance of the WuRx, along with a description of the measurement environment, is often lacking. This absence of detailed information makes it challenging to investigate and replicate these solutions effectively. In this article, we conducted an in-depth examination of the impact of AS3933 configurations. We carried out measurements to investigate the LF performance and behavior of the AS3933. By implementing the improvements identified through LF analysis, we proposed a reliable WuRx circuit designed for operation in the 868 MHz band. This WuRx is capable of reliably detecting signals with an average power as low as −61.6 dBm, while consuming less than 6 μW. Furthermore, we managed to reduce the transmission duration of the WuPt to less than 10 ms.

In Section 2, we present the previous work related to WuRxs based on LFPM and identify research gaps. In Section 3, we describe the measuring methods used. In Section 4, we summarize the investigations conducted and present the outcomes of both the LF and RF analyses. In Section 5, we apply the improvements identified in the previous section and introduce our proposed WuRx designs. The results of the whole article are summarized in Section 6. In Section 7, these results are discussed.

## 2. Previous Work

### 2.1. State of Research

In our analysis, we examined a total of ten publications proposing WuRx designs with LFPM. All implementations are described in detail in the following section. We describe the advantages and disadvantages to show the remaining research gap. In Table 1, the key parameters of the proposed WuRxs are presented. The first two columns correspond to our proposed WuRxs, which are presented in Section 5. The first three rows highlight the primary WuRx performance parameters: MDS, power consumption, and WuPt duration. As highlighted in Table 1, our proposed WuRxs show superior MDS, lower power consumption, and shorter WuPt duration. Subsequent rows provide information on the envelope detector’s properties: RF, detector configuration, and utilized detector diodes. The LF amplifier is characterized in the following rows. In the last five rows, details about the LFPM and its most important parameters are presented. The ON/OFF mode, an integrated duty-cycling option of the AS3933 LFPM to reduce power consumption (see Section 4.1 and [13]), is also included. It is worth noting that we sorted Table 1 by MDS, although the following text is presented in chronological order.

Reference [2] were the first to use aLFPM along with an RF envelope detector. They used the AS3932 as the LFPM. The AS3932, however, only functions in the LF band at 125 kHz. The WuPt employed had a duration of 13 ms and did not employ Manchester coding. The data rate was 2.97 kbit/s per second, and the LF signal operated at 125 kHz, resulting in approximately 42 carrier cycles for each data bit. The only active component in their WuRx was the AS3932, drawing a current of 2.6 μA, indicating that the LFPM’s ON/OFF mode was not used. The MDS was measured using an attenuator setup, while there was no mention of packet error rate (PER) in their description.

Reference [21] proposes two different WuRx designs with two different envelope detectors. The first design, using the Schottky diode HSMS-2822, achieves an MDS of only −31 dBm, while the second design with the HSMS-285C reaches −43 dBm. The MDS values are based on the peak power, resulting in a roughly 6 dB difference. When considering the average power, estimated MDS values are approximately −37 dBm and −50 dBm. The differences between the average and peak power are discussed in the first paragraphs of Section 3. The AS3930 is employed as the LFPM, which is essentially a single-channel version of the AS3932. We calculated the WuPt duration using the provided values in the reference.

Reference [19] introduces two different WuRx designs, one featuring a biased envelope detector and the other an unbiased one. The biased envelope detector exhibits a superior MDS value, but it comes at the cost of increased power consumption. This publication primarily concentrates on the envelope detector, without further investigations on the WuPt and LFPM. Specific values for the LF and addressing schema are not provided. The PER of the WuRx is measured in relation to the distance between the transmitter and receiver.

Reference [17] presents an improved version of the design introduced in [2], now operating in the 433 MHz range. It achieves performance similar to that at a carrier frequency of 868 MHz, with a slight improvement in MDS.

Reference [15] presents one of the first WuRxs employing the AS3933 with a carrier frequency of around 18 kHz. The MDS is improved, thanks to the introduction of a LF amplifier. LF amplifiers based on operational amplifiers (OAs), capable of amplifying an 18 kHz signal, typically have a high power consumption. To address this issue, they introduce an intermediate-frequency logic controller (IFLC). The IFLC activates the LF amplifier only when an RF carrier is detected. It utilizes ultra-low-power OAs and a comparator that monitors the output voltage of the envelope detector. They achieve a PER of 1% measured through an attenuator-based setup. While they provide various information about the WuPt, they do not investigate the configuration of the AS3933. It is worth noting that the IFLC’s performance significantly affects the WuRx’s MDS and power consumption. If the IFLC’s threshold is set too high, the WuRx’s MDS degrades. If it is set too low, the IFLC may falsely trigger, substantially increasing the WuRx’s power consumption. Due to parameter variations in COTS components, we estimate that threshold calibration may be necessary for each prototype. Additionally, this solution could result in rapid battery depletion up to factor 100, particularly in noisy environments.

Reference [14] presents a WuRx capable of receiving frequency modulation (FM) signals. Two envelope detectors are tuned to frequencies of 838 MHz and 868 MHz. Both LF signals are combined in a differential amplifier. A broad frequency deviation of 30 MHz is necessary due to wide band-pass behavior of the envelope detector, resulting in higher power consumption because of the differential amplifier and dual bias voltages. They investigated how different wake-up frequencies affect the MDS. To implement this WuRx in a wireless sensor node, two antennas or an RF coupler are required. Likewise, for the WuTx, two RF transmitters are likely needed to achieve the necessary 30 MHz frequency deviation. It is important to consider legal requirements when transmitting on the 838 MHz frequency.

Reference [16] concentrates on enhancing the MDS through antenna diversity. The combination of signals from different antenna paths occurs after the envelope detector. The authors demonstrate an MDS improvement of approximately 3 dB compared to their design without antenna diversity. However, the article does not investigate the configuration of the LFPM concerning LF, data rate, or addressing.

Reference [10] enhances the WuRx initially introduced in [15]. The primary modification is an extended WuPt duration of 25 ms, due to Manchester coding.

Reference [20] introduces an FM WuRx operating in the 2.45 GHz band. Two envelope detectors equipped with RF filters are employed to demodulate the 2.404 GHz and 2.474 GHz signals. A low-power OA is used to subtract and amplify these signals. The article provides detailed insights into how the WuPt is generated using two Bluetooth Low Energy transmitters. The reduced MDS compared to [19], operating in the same frequency band, could be due to additional losses from the RF filters or cross-talk between both channels. However, using two antennas and two transmitters is not practical when implementing this WuRx in a wireless sensor node.

Reference [18] presents a WuRx operating in the 2.45 GHz band, utilizing IEEE 802.11 packets. This design incorporates a longer WuPt compared to other references. However, the article lacks information on addressing, data rate, methods for MDS measurement, and PER.

### 2.2. Proposed Improvements

An LF frequency exceeding 100 kHz, as employed in prior works like [2,16,17,19,21] results in a high bandwidth and substantial modulation complexity. In our proposed WuRx, we configured the AS3933 to operate in frequency band 5, utilizing an LF of only 25.7 kHz. This reduces the bandwidth by a factor of 4. Moreover, it reduces the modulation complexity by requiring fewer LF carrier pulses per bit. Less than 100 Bytes are needed to generate the WuPt with a typical RF transmitter (see Section 4.4).

We showed various methods to measure the MDS of the WuRx. The methods are summarized in Section 3. To our knowledge, no one in the state of research has discussed their accuracy. In our publication, we measured the PER against the RF input power. We discuss its accuracy in Section 3.2.

References [10,15] likely use the ON/OFF mode of the AS3933 at 11%. Our tests show low duty cycle causes packet loss (Section 4.2.6). The WuRx reliability was not checked in those articles. We tested our design’s reliability by multiple measurements (see Section 5).

The input offset voltage of COTS OAs and comparators affects the MDS of the implementations in the articles [10,14,15,20]. Because components vary, the threshold must be calibrated manually, and noisy environments can drain the battery quickly. Our WuRx does not require an LF amplifier to reach an MDS of −61.9 dBm. By omitting the LF amplifier, we have made the design simpler and improved reliability.

Many designs rely on components that are not commercially available. This, coupled with crucial missing details and values, leads to WuRx designs that cannot be reproduced or compared. We exclusively used COTS components and provided the design parameters.

In Section 4.1, we discuss the operational principles of the AS3933 and its configuration. The results of the LF analysis are detailed in Section 4.2. To construct a WuRx comparable to the current state of research, we integrated an RF envelope detector, as explained in Section 4.3. Further measures to reduce the WuPt duration and minimize the WuRx’s power consumption are outlined in Section 5. With the implementation of these measures, our proposed WuRx attains an MDS of −61.6 dBm, power consumption of 5.71 μW, and WuPt duration of 9.02 ms. The improvements made in comparison to the state of research are evident in Table 1 and are visualized in Section 5.3.

## 3. Proposed Measurement Methods

In many WuRx publications, the sensitivity or MDS is presented, but the definition of MDS and the measurement method are often unclear. In this article, we adopted the following definition, which is adapted from [22]: The receiver MDS is defined as the minimum amount of signal power required at the input of a receiver that results in a certain PER.

Due to OOK modulation, the signal envelope does not remain constant, resulting in unequal peak and average power for the WuPt. The RF WuPt, as discussed in Section 4.4, has a duty cycle of 0.28, leading to a 5.46 dB ratio between the peak and average power. For this publication, we adopt the MDS definition based on average power. Our measurement setup, as detailed in the following subsections, sets the level of the frequency generator equal to the peak power. Consequently, the MDS values presented in the RF measurements in Section 4.5 and Section 5 have been adjusted to account for the duty cycle of the WuPt.

This article focuses on comparing various circuits and setups. However, it is important to note that some methods used in prior research have certain problems. These methods lack the necessary accuracy for a precise comparison between different configurations.

The range measurements conducted in [2] exhibit inaccuracies attributed to factors such as variable antenna gain, multipath fading, or temporary interferers. It is important to note that measurements using the RF generator described in [2] only confirm the carrier burst of the WuPt, without taking into account the preamble and address matching. In the case of [15], a variable attenuator was utilized between the WuTx and WuRx. However, this setup may introduce inaccuracies due to variations in transmission power scattering of the RF signal.

### 3.1. Packet Error Rate Measurement System

The proposed PER measurement system is effective in consistently and repeatedly measuring the MDS for comparing various WuRx designs. Figure 3 shows the block diagram of the PER measurement system.

The central component of the system involves a microcontroller responsible for generating the LF component of the complete WuPt. This LF signal is then fed into the frequency generator. The SML02 frequency generator from Rhode & Schwarz is capable of modulating the LF signal through external pulse modulation. Additionally, the microcontroller can connect to the LFPM through serial peripheral interface (SPI).

PER measurements were performed by counting the number of transmitted packets $n_{\mathrm{TX}}$ and received packets $n_{\mathrm{RX}}$. Equation (Equation 1) shows the estimation of the PER.
(1)$$\mathrm{PER}=1-\frac{n_{\mathrm{RX}}}{n_{\mathrm{TX}}}$$

### 3.2. Minimum Detectable Signal Measurements

We determined the WuRx MDS by adjusting the RF power of the frequency generator. The PC operating the graphical user interface automated this measurement by sending commands to the frequency generator via RS232. The outcome of this measurement is a graph showing the relationship between the RF power and the corresponding PER. An example is seen in Section 4.5.

According to the provided definition, an MDS measurement must define the corresponding PER. We selected a value of 30% for PER because it represents a typical collision probability in a busy WSN environment. We estimated that this collision probability of 30% is reached when 120 sensor nodes transmit RF packets randomly with a duty cycle $p_{\mathrm{TX}}$ of 1:1000. Equation (Equation 2) demonstrates the calculation of the minimum number of network participants *n* required to achieve a collision probability $p_{col}$.
(2)$$n\geq \frac{\log(1-p_{\mathrm{col}})}{\log(1-3\;\cdot \;p_{\mathrm{TX}})}$$

The factor of 3 in the equation accounts for unsynchronized communication. It reflects the increased collision probability between two participants, which is increased by a factor of 3 due to the random overlap of transmitted packets.

The accuracy of the MDS measurement relies on the level precision of the frequency generator SML02. The specification gives a level accuracy of ±0.5 dB in the relevant frequency range following a 15-min warm-up period [23].

Furthermore, there is a random error attributed to the confidence level of the PER estimation. The number of packets sent per PER measurement was configured to be $n_{TX}=100$. Equation (Equation 3) illustrates the calculation of the confidence level *e* using a standard deviation *z* of 2 and the PER.
(3)$$e=z\;\cdot \;\sqrt{\frac{\mathrm{PER}\;\cdot \;(1-\mathrm{PER})}{n_{\mathrm{TX}}}}=2\;\cdot \;\sqrt{\frac{0.3\;\cdot \;0.7}{100}}=9.1\%$$

To translate this confidence level of the PER into the MDS accuracy, we applied the typical relationship between PER and RF power, as shown in Section 4.5. In this graph, the transition from PER=0% to PER=100% occurred within a 1 dB range of RF power. For a confidence level of e=9.1%, this corresponds to an error of roughly 0.1 dB. Therefore, the overall accuracy of the MDS measurement is approximately ±0.6 dB.

## 4. Experimental Analysis of the Low-Frequency Pattern Matcher

In the first subsection, we describe the working principle of the LFPM. In Section 4.2, we present the results of the LF analysis.

### 4.1. Low-Frequency Pattern Matcher

The AS3933 is an LFPM capable of receiving and decoding WuPts within the LF frequency range of 15–150 kHz. It includes a 16-symbol or 32-symbol Manchester pattern correlator. The typical wake-up sensitivity is 80 μV${}_{\mathrm{RMS}}$ or 226 μV${}_{\mathrm{PP}}$ for a sinusoidal input signal. It can operate with a supply current as low as 2.3 μA when supplied with a voltage between 2.4–3.6 V. The AS3933 has three LF channels for 3D pattern detection and localization, although this feature is not the primary focus of WuRx and is disregarded for simplicity in our analysis [13].

The information in the following Section 4.1.1 and Section 4.1.2 is taken from the datasheet [13].

#### 4.1.1. AS3933 Wake-Up Packet

Figure 4 shows the AS3933 WuPt utilized in our LF analysis. The top plot displays the entire packet, while the lower plot zooms in on a portion of the preamble. We employed a carrier frequency of 18.7 kHz and utilized a 32-symbol Manchester pattern.

The AS3933 WuPt consists of three components: the carrier burst, preamble, and pattern. The carrier burst is a continuous carrier signal seen over a period from t=0 to 3.4 ms. The preamble consists of multiple carrier pulses, each with a duration corresponding to the data rate. In this instance, the data rate is set at 2.34 kbit/s, and the preamble is visible from t=3.4 ms to 8.2 ms. The pattern contains a 16-bit address, resulting in a 32-symbol pattern due to Manchester encoding.

#### 4.1.2. AS3933 Block Diagram

The AS3933 is configured using the SPI interface. There are 20 internal registers that can be written to or read from via SPI. These registers influence the AS3933’s operation. In Figure 5, a subset of the register bits are highlighted in gray. The figure shows the simplified block diagram of the AS3933.

The AS3933 receives the LF signal through its LF1P input pin, which corresponds to channel 1 (channels 2 and 3 are not displayed in Figure 5). This signal is then passed to the channel amplifier. We enabled an additional 3 dB of gain with G_BOOST set to 1. The amplification level of the channel amplifier is controlled by the automatic gain control (AGC). Following the reception of a WuPt, the AGC gain can be accessed via SPI and serves as a received signal strength indicator (RSSI). Several SPI registers are available for adjusting the AGC behavior, particularly beneficial in noisy environments. However, these registers are not shown in Figure 5 and will not be explored further in this article.

The amplified LF signal is fed into the frequency detector within the AS3933. By configuring BAND_SEL, one of the five frequency bands available in the AS3933 is chosen. The LF carrier $f_{\mathrm{LF}}$ is determined by the clock generator’s frequency $f_{\mathrm{clk}}$. When $f_{\mathrm{clk}}$ is set to 32.768 kHz, one of the carrier frequencies in the range of 18.72–131.1 kHz is selected. For our investigations, we opted for the lowest frequency band—band 5—which has a carrier frequency of 18.72 kHz. The AS3933 performs continuous frequency detection by counting the number of zero-crossings during a specified time interval. The tolerance for this frequency detection is adjusted by S_WU1. We chose the most relaxed setting with S_WU1 = 0. When an LF signal with the correct carrier frequency is detected, the demodulator is activated, resulting in a minor increase in supply current, ranging from 8.3–12 μA.

The demodulator within the AS3933 produces two output signals: the slow envelope and the fast envelope. These two signals are analyzed by the data slicer to generate the digital bit stream. Adjusting FS_ENV modifies the time constant of the fast envelope signal, while FS_SCL alters the time constant of the slow envelope signal.

The data slicer compares the two envelope signals, incorporating hysteresis for improved noise immunity. By setting HY_20m = 1, the hysteresis level is reduced from 40 mV to 20 mV. With HY_POS = 1, hysteresis is applied exclusively to the positive edge. Instead of the slow envelope signal, an absolute reference signal is selected using ABS_HY = 1. This reference signal is further reduced by S_ABSH = 1. In our configuration, we maintained the default hysteresis settings with all four register values set to zero.

The Manchester decoder generates the decoded bit stream, which is made available on the DAT output pin along with a reconstructed clock signal on CL_DAT. This enables a microcontroller to monitor these pins and read the address of the wake-up packet as well as the subsequent data, allowing for custom addressing or data transmission implementations.

The AS3933 provides several addressing possibilities, including 16-symbol and 32-symbol patterns with and without Manchester coding. The address is stored in the registers TS1 and TS2. T_HBIT is used to define the bit rate of the wake-up packet. In our investigations, we opted for a data rate of 2.34 kbit/s to ensure a whole number of carrier periods per bit, as explained in Section 4.4. As our default configuration, we enabled Manchester coding and used the 16-bit address 0x5C2F.

The AS3933 can operate in the ON/OFF mode or duty cycling mode to reduce its power consumption. In our configuration, we used ON_OFF = 0, meaning we did not employ duty cycling. However, the off-time of the AS3933 can be adjusted by changing T_OFF, with the possible values of 1, 2, 4, and 8 ms. The on-time is fixed at 1 ms. This results in duty cycles of 50%, 33%, 20%, and 11%, respectively.

The AS3933 clock generator can be driven by three different sources: an external clock, a crystal oscillator, or an RC oscillator. The first two sources do not require calibration, but the RC oscillator must be calibrated using the SPI communication. The RC oscillator is commonly used in the state of research because it gives the lowest power consumption. The RC oscillator frequency can be calibrated within the range of 25–45 kHz. For our investigations, we calibrated the RC oscillator of the AS3933 to 32.768 kHz, which is a frequency that can be accurately generated from a clock crystal commonly used in low-power wireless sensor nodes.

#### 4.1.3. State-of-the-Art Profile

Table 2 shows a detailed list of the AS3933 and WuPt configurations. These values were designed based on existing research and the AS3933 datasheet [13]. We selected band 5 with a carrier frequency of 18.72 kHz to address the issues discussed in Section 2.1 (high LF leading to high RF bandwidth and WuPt complexity). These configuration values serve as the baseline for our subsequent investigations. Analyzing the effects of changes in all registers simultaneously is not feasible due to the large number of registers, values, and external parameters (see Figure 5). Therefore, we examined the effects of one or two parameters at a time. We refer to the baseline values presented in Table 2 as the state-of-the-art profile (SoAP). The SoAP is highlighted in the following investigation results with a star symbol (*). The WuPt corresponding to the SoAP was shown in Figure 4.

### 4.2. Low-Frequency Analysis

#### 4.2.1. Low-Frequency Measurements

We generated the LF signal using the frequency generator and directly fed it into the AS3933, along with a 50 Ω termination resistor. In Section 3.2, we explained our methodology for measuring the RF MDS. We applied a similar approach for LF MDS measurement, but used voltage levels ($\mu V_{\mathrm{PP}}$) instead of power levels (dBm). The conversion between these values is determined by the 50 Ω voltage drop.

Figure 6 shows the measurement results for the PER and the RSSI reading from the AS3933 in relation to the peak-to-peak signal level of the WuPt. The MDS was 138 μV${}_{\mathrm{PP}}$ at a PER of 30%. This indicates an improved MDS compared to the values provided in the datasheet, which are 80 μV${}_{\mathrm{RMS}}$ or 226 μV${}_{\mathrm{PP}}$. The AS3933 RSSI reading exhibited a linear relationship within its operational range. The RSSI value is constrained to integer values ranging from 0–31. At a 3 V supply voltage, the AS3933 consumed 3.30 μA, resulting in a power consumption of 9.90 μW.

#### 4.2.2. Frequency Bands and Data Rate

We conducted an investigation to see how different frequency bands and data rates affect the AS3933’s signal detection. For each carrier frequency, we calculated the lengths of the carrier burst, preamble, and addressing segments as specified in the datasheet [13]. Additionally, we adjusted the time constant of the fast envelope based on the chosen data rate. Table 3 shows the settings that were changed in comparison to the SoAP. We chose data rates with $f_{\mathrm{LF}}/f_{\mathrm{bit}}$ as a multiple of 4, as explained in Section 4.4. The last column provides the measured MDS, determined using the PER measurement system with a 30% PER.

In the second row, we have the SoAP. We tested other configurations with a modulation ratio of 8 in higher frequency bands, but they exhibited a slightly lower MDS. Among these configurations, the band 2 configuration offered the best MDS. However, it is important to note that using band 2 is not practical due to its high RF bandwidth and increased modulation complexity.

#### 4.2.3. Data Slicer and Preamble Length

The datasheet [13] presents that both the fast and slow envelope signals are input into the data slicer. The data slicer functions as a comparator, producing the digital data signal. This comparator can be configured with adjustable hysteresis settings to enhance noise immunity. When modifying the slow envelope, it becomes necessary to extend the preamble’s length correspondingly. This ensures the proper turn-on time constant for the slow envelope and the correct reception of the WuPt. Transitioning to the absolute data slicer configuration was expected to yield additional improvements. It might be possible to decrease the WuPt duration by omitting the preamble. Table 4 shows the impact of different data slicer configurations on the AS3933’s MDS level.

The hysteresis level settings, either 20 mV or 40 mV, had minimal impact on the MDS. Enabling hysteresis exclusively on the positive edge significantly worsened the MDS. Utilizing absolute hysteresis slightly degraded the MDS, whereas employing the reduced absolute hysteresis setting led to a minor improvement in the MDS.

#### 4.2.4. Deviation of Low Frequency and Data Rate

The RC oscillator frequency of the AS3933 impacts both the LF and data rate. Calibration allows adjustment within the range of 25–45 kHz. In our first measurement, we modified the RC oscillator calibration, resulting in either accelerated or decelerated WuPts. This provides the possibility for using shorter WuPts without altering the overall AS3933 configuration or switching to a different frequency band. Table 5 shows the carrier frequency, data rate, WuPt duration, and the corresponding MDS for various RC oscillator calibration frequencies. The MDS showed slight improvement with higher RC oscillator frequencies but degraded when set to a very low frequency of 25 kHz.

We examined the impact of altering the carrier frequency of the WuPt while keeping the data rate and AS3933 clock frequency unchanged. This investigation aimed to assess the effects of distortions or interference from other RF systems on the signal detection. As specified in [13], the AS3933’s frequency detector matches the carrier frequency. The register S_WU1 determines the tolerance of this frequency detection. S_WU1 offers possible values ranging from 0—most relaxed to 2—tightest tolerance setting.

Figure 7 shows the MDS levels for various carrier frequencies and the three tolerance settings. With S_WU1 = 0, the AS3933 successfully detected WuPts within the frequency range of 9–70 kHz, with the lower limit determined by the frequency generator’s capabilities. When S_WU1 = 1, the AS3933 detected WuPts within 10–67 kHz, and for S_WU1 = 2, the range was 12–50 kHz. The MDS at the calculated carrier frequency of 18.7 kHz was slightly worse with tighter tolerance settings. The optimal MDS was not achieved at the theoretically expected carrier frequency but rather at a slightly increased frequency of approximately 25 kHz. These findings suggest that the AS3933’s frequency detector may not effectively filter the carrier frequency. Consequently, it might incorrectly detect RF packets from other systems, resulting in increased power consumption due to false wake-ups. The AS3933 heavily relies on proper filtering of the analog input signal.

In our next measurement, we kept the AS3933 configuration constant and only altered the data rate of the WuPt. We conducted these investigations to demonstrate the potential effects of interference from other RF systems or a misaligned calibration of the RC oscillator. Figure 8 shows the impact of data rate variations on the MDS. It is evident that the AS3933 is sensitive to changes in data rate. It successfully detected signals within a range of 2.2–2.6 kbit/s, which corresponds to a relative deviation of approximately 11%. These findings suggest a robust signal filtering for pattern detection in the AS3933. It is unlikely that false detections of the AS3933 pattern will occur.

#### 4.2.5. Pattern Correlator

When we modified the address pattern and the AS3933 configuration to use 8-bit Manchester coding, the duration of the WuPt was naturally reduced. In the case of the SoAP, the WuPt duration was minimized to 15.0 ms. The measured MDS remained constant at 138 μV${}_{\mathrm{PP}}$.

#### 4.2.6. Power Saving Modes

To further reduce the power consumption of the AS3933, we examined three power-saving measures: disabling the gain boost (G_BOOST), enabling the ON/OFF mode, and reducing the supply voltage.

When we disabled the gain boost, the MDS increased as expected. With gain boost, we measured an MDS of 138 μV${}_{\mathrm{PP}}$, while without gain boost, it increased to 195 μV${}_{\mathrm{PP}}$. This measured difference closely matched the 3 dB difference presented in the datasheet. Regarding the supply current, we measured 3.30 μA with gain boost enabled and 3.22 μA without gain boost. The measurements suggests that the AS3933 did not achieve the specified 150 nA reduction as noted in its datasheet [13].

Enabling the ON/OFF mode of the AS3933 leads to packet loss due to random misses in the carrier burst of the WuPt. To mitigate this issue, increasing the carrier burst is necessary. Figure 9 shows the PER curves for different $t_{\mathrm{off}}$ values, while maintaining a constant carrier burst length of 3.42 ms. As observed, random packet loss and a degradation in the MDS occurred. However, with $t_{\mathrm{off}}=1\;\mathrm{ms}$, no packet loss was experienced.

Extending the carrier burst length $t_{\mathrm{CB}}$ by 1,2, and 4 ms reduced random packet loss. However, attempting to further extend $t_{\mathrm{CB}}$ beyond 7.26 ms was not feasible, as the AS3933 could not reliably receive packets under these conditions. Consequently, we were unable to measure the MDS because the PER did not fall below 30%. An overview of these measurements is provided in Table 6.

Extending the carrier burst length resulted in a noteworthy reduction in the AS3933’s supply current. However, it was unable to completely eliminate random packet loss, even at higher $t_{\mathrm{off}}$ values.

To reduce power consumption further, we examined the impact of lowering the supply voltage. We tested various supply voltages and measured the supply current along with the MDS. We measured down to 2.4 V as per the AS3933 datasheet [13]. Table 7 shows the measurement results. The supply current decreased significantly, with only a minor drop in the MDS.

#### 4.2.7. AS3933 Input Impedance

The AS3933’s input impedance is a crucial parameter, particularly when integrating a passive input filter or an LF amplifier circuit. The datasheet [13] specifies a typical input impedance of 2 MΩ at 125 kHz. To verify this value, we conducted the following measurement: We introduced a series resistor $R_{s}$ between the output of the frequency generator and the input of the AS3933. As a result, the input voltage at the AS3933 decreased due to the voltage drop across $R_{s}$. Measuring this voltage drop directly with an oscilloscope was not feasible due to the typical probe impedance of 10 MΩ and probe capacitance of several picofarads. Instead, we utilized the AS3933’s RSSI reading to estimate the AS3933’s input resistance.

We estimated the AS3933’s reference level, which is the theoretical input voltage corresponding to a 0 dB RSSI value (see Figure 6), using linear regression. Based on the change of the reference level with $R_{s}$ and without $R_{s}$, we can estimate the voltage drop and the input resistance. In the final measurement, we selected a series resistor close to $R_{\mathrm{in}}$ to minimize the influence of measurement errors. This measurement was conducted in both carrier frequency bands 1 and 5. The results of the input impedance measurement are summarized in Table 8.

The measurements revealed a significant discrepancy from the input impedance value provided in the datasheet. According to the measurements, the AS3933’s input impedance was more than one order of magnitude lower than what was specified in the datasheet.

### 4.3. Passive Envelope Detector

Using LF signal transmission within a WSN faces practical challenges:Large transmitter and receiver coil needed.High transmitter power consumption during WuPt transmission, especially problematic for battery-powered WuTx.Limited transmission range, only a few meters.

To address these issues, transmitting the WuPt modulated in RF is a more practical option. However, this approach requires a low-power or passive RF-to-LF converter. Many COTS WuRxs use envelope detectors based on Schottky diodes. These detectors are suitable for demodulating simple amplitude-modulated signals or OOK. Envelope detectors rely on nonlinear components to convert the signal. After this conversion, a low-pass filter is applied to isolate the LF envelope, removing the RF components.

In a previous study, we examined the passive envelope detector employing zero-biased Schottky diodes, as detailed in [24]. Here, we provide a summary of the findings from this investigation.

The envelope detector is described by the open-circuit output voltage $V_{\mathrm{out}}^{\mathrm{OC}}$ as a function of the incident RF power $P_{\mathrm{in}}$. For low input power, the output voltage is proportional to the input power. Equation (Equation 4) defines the open-circuit voltage sensitivity $\gamma_{\mathrm{OC}}$ [4] (p. 568).
(4)$$V_{\mathrm{out}}^{\mathrm{OC}}=\gamma^{\mathrm{OC}}\;\cdot \;P_{\mathrm{in}}$$

The diodes behavior is modeled as a Thévenin equivalent, with the open-circuit voltage $V_{\mathrm{out}}^{\mathrm{OC}}$. The equivalent resistance is known as the video resistance $R_{v}$ [4] (p. 569).

For our investigations in the RF range, we employed a voltage-doubler circuit with impedance matching using lumped components. The circuit is schematic is shown in Figure 10. Impedance matching is achieved through components $L_{1}$ and $C_{1}$, while $C_{2}$ serves as a low-pass filter. We selected SMS7630 diodes for this circuit. Our characterization of this circuit, as detailed in [24], revealed a voltage sensitivity of γ=45 mV/μW and a video resistance of 11.7 kΩ.

### 4.4. Wake-Up Packet Generation

We used an 868 MHz OOK signal for transmitting the WuPt. The envelope detector’s function is to convert the RF signal into a valid AS3933 LF WuPt. For this process to work seamlessly, the WuTx needs to be programmed in a manner that ensures the appearance of an AS3933-detectable WuPt on the output of the envelope detector.

Figure 11 shows the step-by-step generation of the WuPt by a commercial RF transmitter. The LF carrier of the AS3933 $f_{\mathrm{LF}}$ is constructed by transmitting a pattern of 10101010 with the RF transmitter. To achieve this, the data rate of the RF transmitter must be twice the carrier frequency of the AS3933. The pattern 10101010 represents four signal periods, and it is crucial that the ratio between the carrier frequency and the data rate of the AS3933 is a multiple of four. This ensures a byte-wise-continuous modulation pattern, eliminating the need for bit-shift operations when preparing the WuPt within the WuTx.

The entire message is programmed into the RF transmitter. Table 9 shows the hexadecimal pattern of an example WuPt in band 5, with a data rate of 2.34 kbit/s. It uses the 16-bit address 0x5C2F and employs Manchester coding. It is important to note that for the subsequent experiments, no RF transmitter is utilized. Instead, as described in Section 3.1, we programmed this bit pattern into the microcontroller of the PER measurement system. The frequency generator then outputs the OOK-modulated RF packet.

### 4.5. Radio-Frequency Analysis

The performance of the WuRx was investigated in the RF domain using the envelope detector detailed in Section 4.3. Figure 12 shows a photograph of this printed circuit board (PCB). This PCB is part of the measurement system, that we presented in Figure 3. The figure shows the RF connector, envelope detector circuit, AS3933, and the SPI connection.

For the subsequent tests, we populated the circuit shown in Figure 13a. Figure 13b shows the simplified LF equivalent circuit, which includes the replacement of voltage doubler (VD) with a constant-voltage source and the video resistance $R_{v}$, as well as the substitution of the AS3933 with its input resistance $R_{\mathrm{in}}$. A passive band-pass filter was created using the filter capacitor $C_{3}$ and the high-pass capacitor $C_{4}$. This analog filter is essential, as demonstrated by the results in Section 4.2.4 and Figure 7.

Based on our previous investigations (see Section 4.3 and Section 4.2.7), we utilized the values $R_{v}=11.7\;k\Omega$ and $R_{\mathrm{in}}=155\;k\Omega$. The capacitor values for $C_{3}$ and $C_{4}$ were chosen to ensure appropriate band-pass characteristics within band 5.

Figure 14 shows the PER and RSSI measurements of the AS3933 in relation to the RF input power. The WuRx successfully received WuPts with a PER of 30% down to an average signal power of −62.0 dBm. The RSSI reading exhibited a linear behavior within the AS3933’s operating range. The configuration used for these measurements was the SoAP presented in Table 2, resulting in a WuPt duration of 21.8 ms and a supply current of 3.30 μA. These measurements demonstrate that the findings from our LF analysis can be applied to an RF WuRx. When we perform calculations based on Equation (Equation 4), it becomes evident that the WuRx’s MDS is purely dependent on the voltage sensitivity of the envelope detector and the LF MDS of the AS3933.

### 4.6. Repeatability of Results

We conducted repeatability tests by replicating several measurements using five additional prototypes. The prototypes, shown in Figure 15 alongside the one presented in Figure 12, include additional components such as microcontrollers, batteries, RF transceivers, and RF switches. These PCBs were assembled using a pick-and-place machine, eliminating the possibility of manual RF front-end tuning. Nevertheless, the WuRx circuit remains identical, allowing for comparable measurements.

We conducted measurements to determine the voltage sensitivity $\gamma^{\mathrm{OC}}$ of the envelope detector and the MDS of the WuRx. Using these values, we estimated the LF MDS of the AS3933 (see Equation (Equation 4)). The results of these measurements and calculations are summarized in Table 10.

The measurements indicate a degradation in both voltage sensitivity and MDS values. However, when we calculated the LF MDS of the AS3933, we observe that these values closely align with the measurements from our LF analysis. This suggests that the WuRx MDS is primarily influenced by the performance of the envelope detector. The degradation in performance can be attributed to the lack of manual matching for the envelope detectors in the five prototypes. It is important to note that this article does not focus on the RF envelope detector, and despite these issues, the prototypes still perform adequately.

## 5. Design Improvements and Experimental Investigations

When designing a WuRx, there is always a trade-off between MDS, power consumption, and latency. In the case of the AS3933-based WuRx, its always-on design and quick response time mean that the latency is equal to the duration of the WuPt. Table 11 summarizes all the parameters of the SoAP, as well as the parameters of the RF WuRx hardware.

### 5.1. Proposed Design Improvements

Based on the LF analysis results, we propose some ways to lower power consumption without significantly affecting signal detection performance. For instance, we excluded disabling the gain boost, as it reduces the MDS by 3 dB (see Section 4.2.6). Table 12 below outlines these suggestions and their associated parameters. We conducted measurements for MDS and power consumption, and calculated the WuPt duration.

(1) The AS3933’s ON/OFF mode reduces its power consumption by switching between listening and standby modes. We previously analyzed this mode’s performance in Section 4.2.6. The datasheet [13] suggests extending the WuPt’s carrier burst to avoid random packet loss caused by the AS3933’s duty-cycling behavior. However, based on our analysis, it appears that extending the carrier burst may not be necessary, as we achieved a power consumption reduction to 7.38 μW with $t_{\mathrm{off}}=1\;ms$.

(2) As presented in Table 6, the settings of $t_{\mathrm{off}}=4\;ms$ and 8 ms are not practical due to random packet loss, even with longer carrier burst durations. However, with $t_{\mathrm{off}}=2\;ms$, the power consumption was reduced to 6.48 μW. It is worth noting that a carrier burst duration of 5.55 ms is required instead of the previous 3.45 ms when using this setting.

(3) In Section 4.2.6, we explored the reduction of power consumption in the AS3933. According to the datasheet [13], the AS3933 operates reliably down to 2.4 V. When supplied with a lower voltage, the device’s supply current decreases. Consequently, the power consumption of the WuRx was reduced to 7.32 μW at a supply voltage of 2.4 V.

We propose several enhancements to reduce the WuPt duration without significantly affecting the MDS. As a result, we excluded the findings from Section 4.2.2 since all the measures examined in that section deteriorated the MDS.

(4) The AS3933’s data slicer is a comparator, generating the digital bit stream for pattern matching. Our analysis of the data slicer settings, as shown in Table 4, reveals that using a absolute reference signal only slightly degrades the MDS. The advantage of employing a absolute reference signal is that it eliminates the need for settling time associated with the variable reference signal. The settling of the variable reference signal is performed during the reception of the WuPt preamble. When using the absolute reference signal, no preamble is required, reducing the WuPt duration to 17.5 ms.

(5) The LF carrier frequency of the WuPt is directly tied to the clock frequency of the AS3933, which is determined by the selected clock source. In our experiments, we opted for the AS3933’s internal RC oscillator due to its lower power consumption. The RC oscillator can be calibrated via SPI within a frequency range of 25–45 kHz. A higher clock frequency results in a higher LF carrier frequency and faster WuPt. With a clock frequency of 45 kHz, the WuPt duration was reduced to 14.0 ms. As observed in the investigations detailed in Table 5, this adjustment led to a slight improvement in MDS and a minor increase in power consumption.

(6) The AS3933 offers two different pattern lengths. In the SoAP, we employed the 32-symbol pattern mode, which allows for a configurable 16-bit address due to Manchester coding. Alternatively, a 16-symbol pattern and an 8-bit address can be utilized. This configuration reduces the WuPt duration to 15.0 ms, with power consumption remaining unchanged. However, the MDS is slightly degraded to −61.8 dBm due to the slightly increased average power of the WuPt (as explained in the initial paragraphs of Section 3). The feasibility of decreasing the address length of the WuRx depends on the specific application requirements.

### 5.2. Combining Design Improvements

As the final step in the proposed WuRx design, we conducted tests with various combinations of the suggested improvements (1)–(6). Not all possible combinations are practical or yield reliable measurement results. For instance, combinations involving improvement (4) resulted in random packet loss. The combinations (1,3,5) and (1,3,5,6) exhibited the best outcomes in terms of power consumption, MDS, and WuPt duration. The choice between these two combinations depends on the required address length. On the other hand, the combination (2,3) exhibited the lowest power consumption. Table 13 shows five potential combinations and specifies the proposed improvements for each.

### 5.3. Comparison with State of Research

Figure 16 and Figure 17 show the performance of each WuRx implementation. All references from Table 1, where the corresponding values are available, are marked in the figures. Additionally, selected profiles from Table 13 are also presented. In each figure, we explore the relationship between two of the parameters: MDS, power consumption, and latency. References presented in the bottom-right of each plot, show the best values in MDS, power consumption, and WuPt duration. Specifically, improvements (2,3) exhibit the lowest power consumption at 4.73 μW, while improvements (1,3,5,6) demonstrate the lowest latency at 9.02 ms among the analyzed state-of-the-art solutions based on LFPMs.

## 6. Summary

### 6.1. Low-Frequency Analysis

In Section 4.2, we conducted an analysis of the LFPM through measurements in the LF domain. The SoAP showed an MDS of 138 μV${}_{\mathrm{PP}}$. The LF analysis revealed the following key insights:Utilizing the absolute threshold as the data slicer reference allows for the complete removal of the preamble in WuPt.Increasing the RC oscillator frequency enables a faster WuPt without compromising the MDS.The internal frequency detector exhibits a wide operating range, which may lead to false wake-ups and increased power consumption in noisy environments.The ON/OFF mode with $t_{\mathrm{off}}=4\;ms$ or 8 ms results in random packet loss, even when the carrier burst is extended as required.The measured input resistance of the LFPM in frequency band 5 was 155 kΩ, which significantly differs from the 2 MΩ value stated in the datasheet [13].

### 6.2. Radio-Frequency Analysis

We built up an RF WuRx using the LFPM in together with a VD-based envelope detector. Our measurements indicated that the MDS of the RF WuRx is primarily determined by the voltage sensitivity of the envelope detector and the MDS of the LFPM. We obtained an MDS of −62.0 dBm with this prototype, surpassing the MDS achieved by any other publication we examined in Section 2.1.

We validated the consistency of our findings by constructing five additional sensor nodes and conducting MDS measurements. The RF envelope detectors, lacking manual matching, led to increased reflection factors, reduced voltage sensitivity, and a 3 dB MDS degradation. However, the calculated MDS of the LFPM remained relatively consistent across all prototypes. Consequently, we can conclude that the MDS of our proposed design is mainly influenced by the envelope detector’s performance. It is worth noting that the 3 dB MDS loss corresponds to only a 30% reduction in range, as per the Friis equation [25].

### 6.3. Proposed Wake-Up Receiver Improvements

Different applications have different requirements for the WuRx. The factors of MDS, power consumption, and latency often entail trade-offs, and the specific balance needed depends on the application’s requirements.

The SoAP achieved an MDS of −62.0 dBm. The AS3933 operated continuously, consuming a total of 9.90 μW. The WuPt utilized an 18.7 kHz carrier frequency and a 16-bit Manchester-coded address, resulting in a latency of 21.8 ms.

The power consumption dropped when we turned on the ON/OFF mode and lowered the AS3933’s supply voltage. This brought the power consumption down to a minimum of 4.73 μW. However, it led to a minor increase in latency to a value of 23.9 ms.

We managed to reduce latency or the WuPt duration by increasing both the data rate and LF and switching to an 8-bit Manchester-coded pattern. This led to a minimum latency of only 9.02 ms. However, power consumption saw a slight increase to 10.2 μW due to the higher AS3933 clock frequency. Importantly, all these measures had only a minor impact on the MDS.

We combined improvements in power consumption and latency into two different profiles, accommodating both 16-bit and 8-bit addressing. These profiles achieved MDS values of −61.9 dBm and −61.6 dBm, respectively, with a power consumption of 5.71 μW for both. The 16-bit addressing profile had a latency of 14.0 ms, while the 8-bit addressing profile achieved a latency of 9.02 ms.

## 7. Discussion

Compared to other references, we chose to analyze the LFPM within its LF range. Our LF analysis highlighted the significance of conducting measurements to validate its properties. Most notably, measuring the input resistance and verifying the ON/OFF mode emphasized the importance of not solely relying on datasheet information. Reflecting on these issues within the reviewed state of research underscores the importance of discussing the configurations and measurement setups of one’s WuRx implementation. The results of our measurements have raised doubts about whether certain publications reliably achieved their claimed parameters. Since our work focused on the LF range, we were able to eliminate certain effects such as noise, RF interference, and dependence on the envelope detector. Furthermore, our findings can be applied to non-RF designs, as demonstrated in [26,27]. These WuRx designs utilize acoustic and optical transmission for the WuPt.

In Section 5.3, we presented a comprehensive comparison of our proposed WuRx with existing LFPM-based WuRxs in the state of research. Our WuRx showed superior performance across the key parameters: MDS, power consumption, and latency. We achieved an MDS of −61.9 dBm. No other LFPM-based WuRx has achieved such an MDS. Our design also excelled in terms of power consumption, achieving a value of 5.71 μW. None of the other LFPM-based implementations reached this level of power efficiency. The second lowest power consumption was presented in the work by [10], and our design represents a 20% improvement. Additionally, by reducing the WuPt to an 8-bit Manchester-coded address, we were able to decrease the WuPt duration to 9.02 ms. Having over 100 possible address combinations is probably enough for a real-world scenario. None of the analyzed implementations achieved a WuPt duration below 10 ms. In comparison, [2,17] achieved the second lowest WuPt duration, making our design 30% faster. Reducing the WuPt duration not only lowers latency but also benefits the power consumption of the WuTx, which is crucial in battery-powered applications, such as multi-hop communications.

When comparing our proposed WuRx to all known COTS implementations, we acknowledge certain deficits in single key parameters. For instance, comparator-based WuRxs, as demonstrated by [5], exhibit the lowest latency and power consumption. However, they have limited MDS values. In contrast, certain implementations prioritize achieving the best possible MDS values. However, this often comes at the cost of latency, with values exceeding 100 ms in some cases, as observed in the works of [6,7,24]. One of the most significant challenges in non-LFPM-based WuRxs is the parameter variations of COTS components. These variations are particularly pronounced in OAs and comparators, where input offset voltage can differ significantly from one component to another. As a result, it leads to varying MDS values or necessitates manual trimming and threshold calibration to achieve desired performance levels.

In contrast, our proposed LFPM-based WuRx does not rely on LF amplifiers to achieve a good MDS value. As confirmed by the repeatability analysis in Section 4.6, our proposed design is highly reliable. The LFPM we utilized demonstrates a stable MDS value. Notably, the MDS of our design depends solely on the voltage sensitivity of the passive envelope detector. This is a common challenge faced by most COTS WuRxs, primarily because they rely on passive envelope detectors based on Schottky diodes. Our proposed design is highly efficient in terms of board size and material costs, requiring only eight discrete surface-mounted devices.

## Figures and Tables

**Figure 1 sensors-23-08188-f001:**
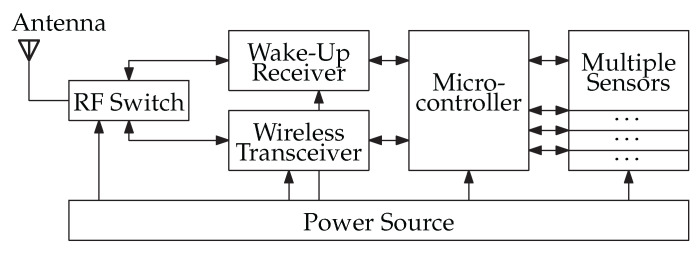
A wireless sensor node with a wake-up receiver (WuRx) according to [2]. An RF switch is used to separate both receiving paths of the WuRx and the wireless transceiver.

**Figure 2 sensors-23-08188-f002:**
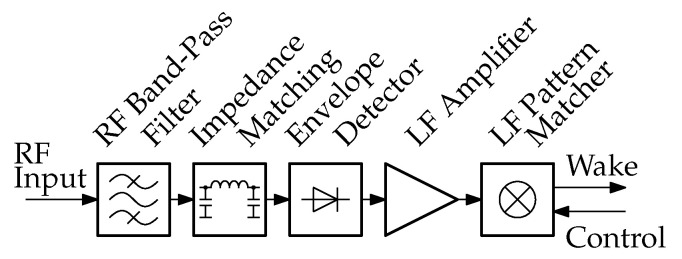
Building blocks of a typical commercial off-the-shelf WuRx with passive envelope detector and low-frequency pattern matcher.

**Figure 3 sensors-23-08188-f003:**
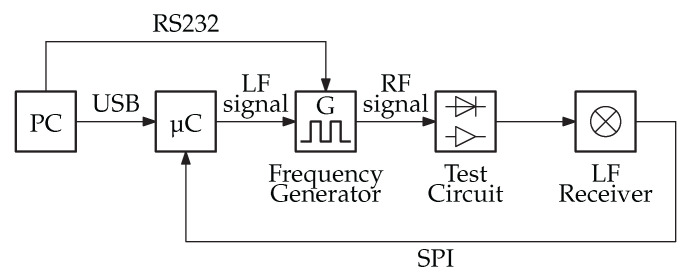
Block diagram of the proposed packet error rate measurement system.

**Figure 4 sensors-23-08188-f004:**
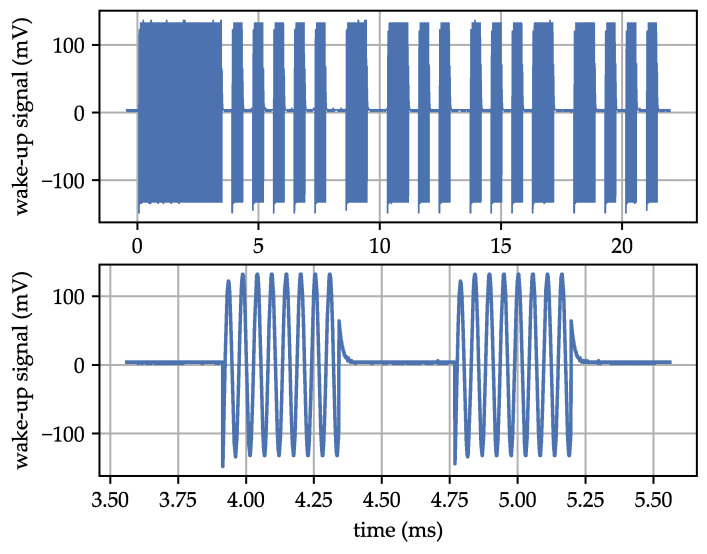
Oscilloscope capture of the utilized AS3933 wake-up packet with a carrier frequency of 18.7 kHz and 32-symbol Manchester pattern.

**Figure 5 sensors-23-08188-f005:**
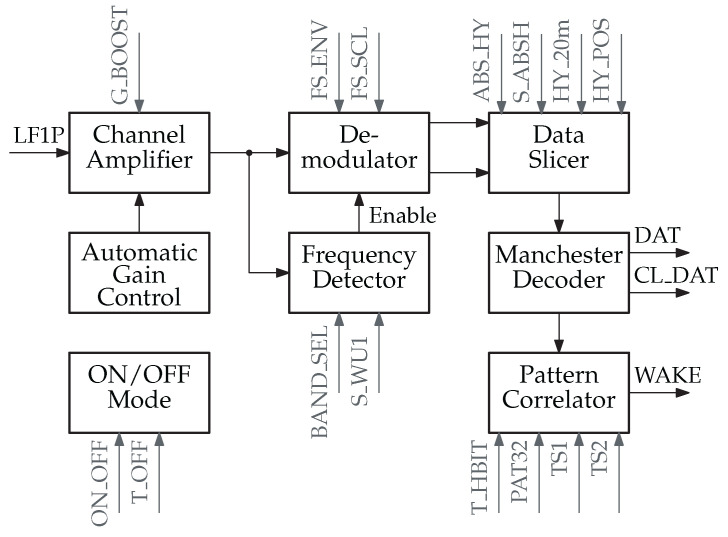
Block diagram of the LF pattern matcher AS3933 according to [13].

**Figure 6 sensors-23-08188-f006:**
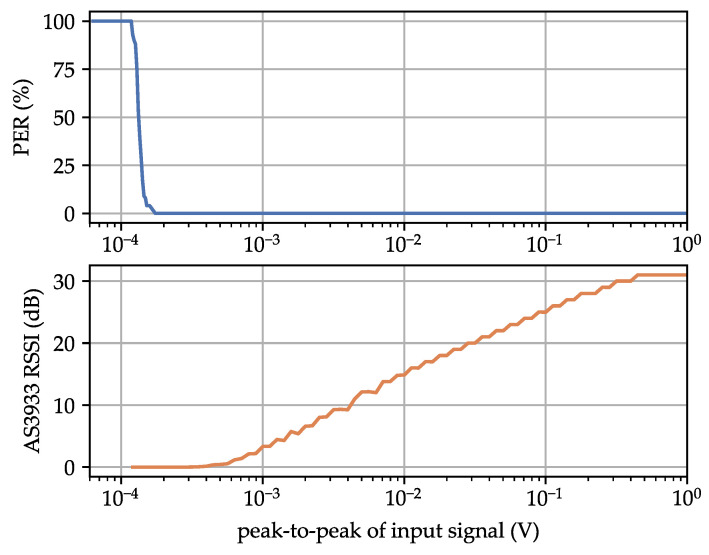
Measurement of the packet error rate and AS3933 received signal strength indicator (RSSI) reading in the relationship to the peak-to-peak signal level of the LF WuPt.

**Figure 7 sensors-23-08188-f007:**
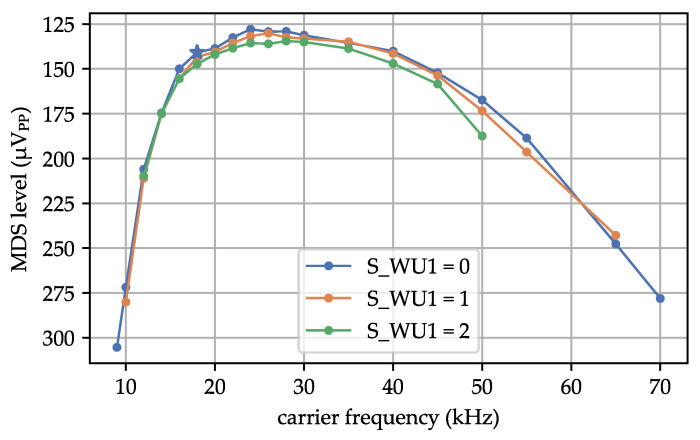
Measurement of MDS of different carrier frequencies and tolerance settings (S_WU1). * state-of-the-art profile.

**Figure 8 sensors-23-08188-f008:**
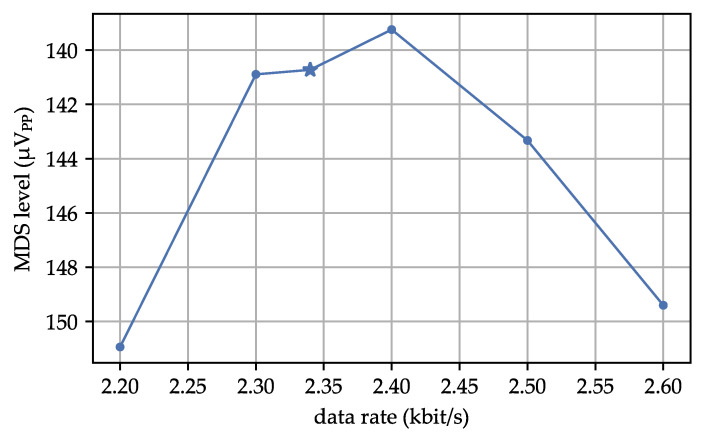
Measurement of MDS of different data rates while AS3933 configuration stays constant. * state-of-the-art profile.

**Figure 9 sensors-23-08188-f009:**
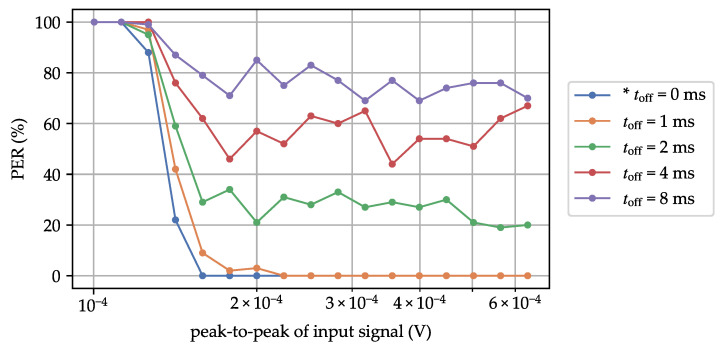
Packet error rate curve for different $t_{\mathrm{off}}$ and constant carrier burst length of 3.42 ms. * state-of-the-art profile.

**Figure 10 sensors-23-08188-f010:**
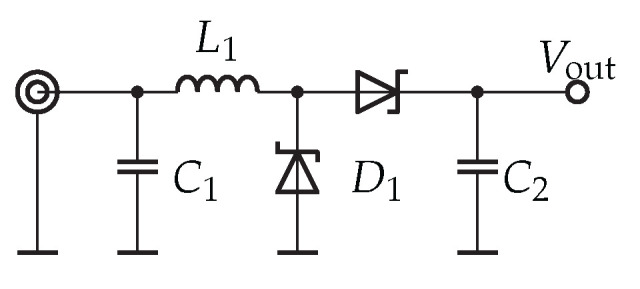
Envelope detector used in this article: a voltage doubler with lumped component impedance matching.

**Figure 11 sensors-23-08188-f011:**
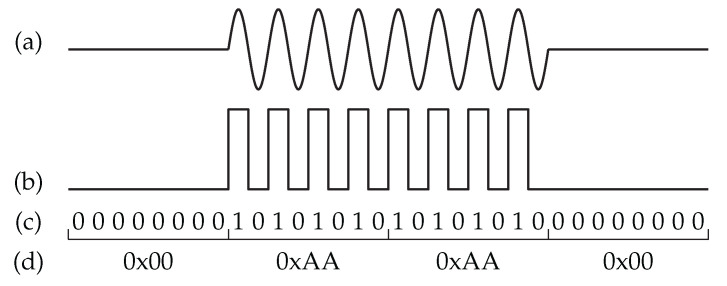
Generation of the wake-up packet. (**a**) Expected WuPt of the AS3933, (**b**) ideal output of the envelope detector, (**c**) wake-up transmitter bit stream, and (**d**) byte-wise presentation of the bit stream.

**Figure 12 sensors-23-08188-f012:**
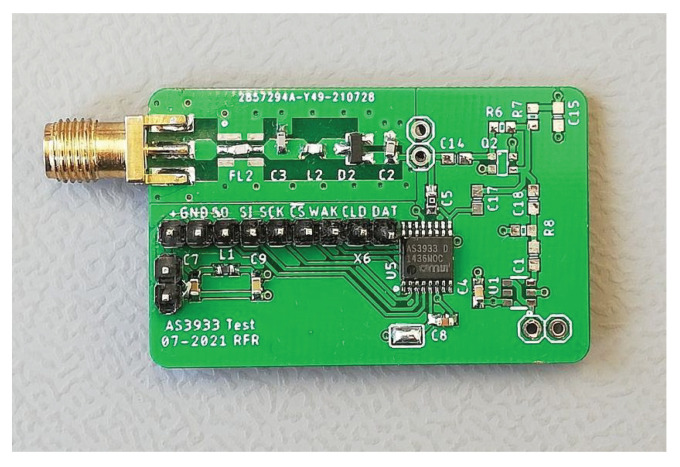
Photograph of the prototype WuRx with voltage doubler envelope detector and AS3933 for 868 MHz.

**Figure 13 sensors-23-08188-f013:**
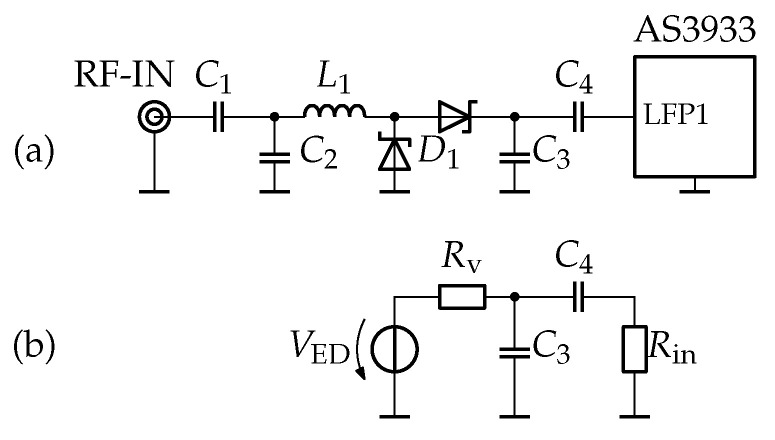
Voltage doubler envelope detector and AS3933. Utilized test circuit in (**a**) and low-frequency equivalent circuit in (**b**).

**Figure 14 sensors-23-08188-f014:**
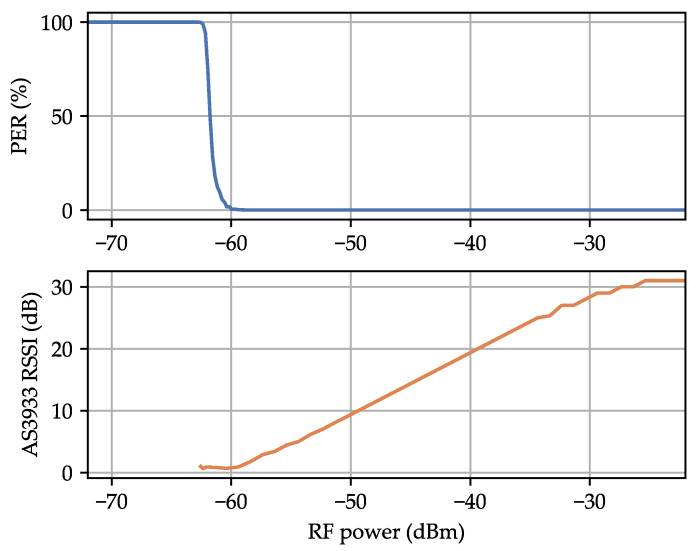
Measurement of the packet error rate and AS3933 received signal strength indicator (RSSI) reading in relation to the RF input power.

**Figure 15 sensors-23-08188-f015:**
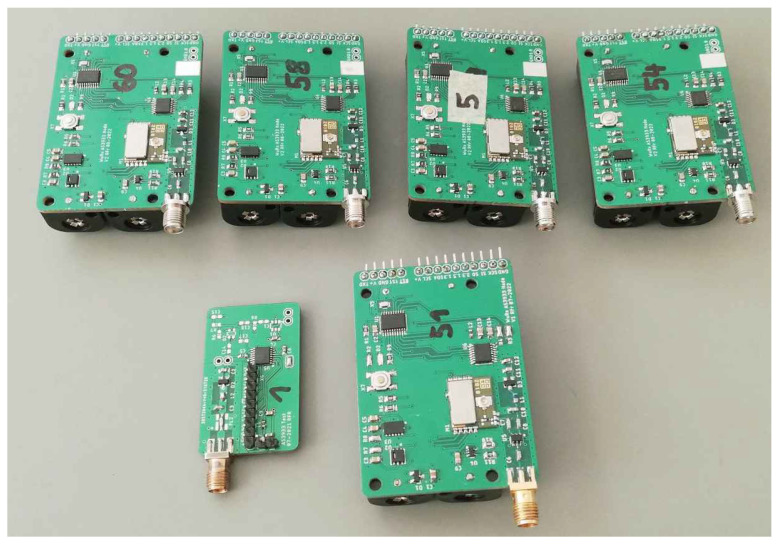
Photograph of multiple prototype WuRxs used to verify the repeatability of our results.

**Figure 16 sensors-23-08188-f016:**
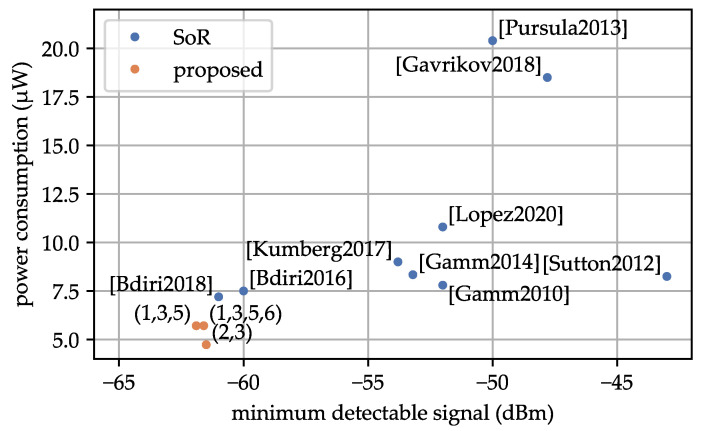
Comparison between MDS and power consumption of the state of research and the proposed WuRx design. The axis displaying the power consumption was limited to 20 μW. References: [2] [Gamm2010], [10] [Bdiri2018], [15] [Bdiri2016], [16] [Kumberg2017], [17] [Gamm2014], [18] [Lopez2020], [19] [Pursula2013], [20] [Gavrikov2018], [21] [Sutton2012].

**Figure 17 sensors-23-08188-f017:**
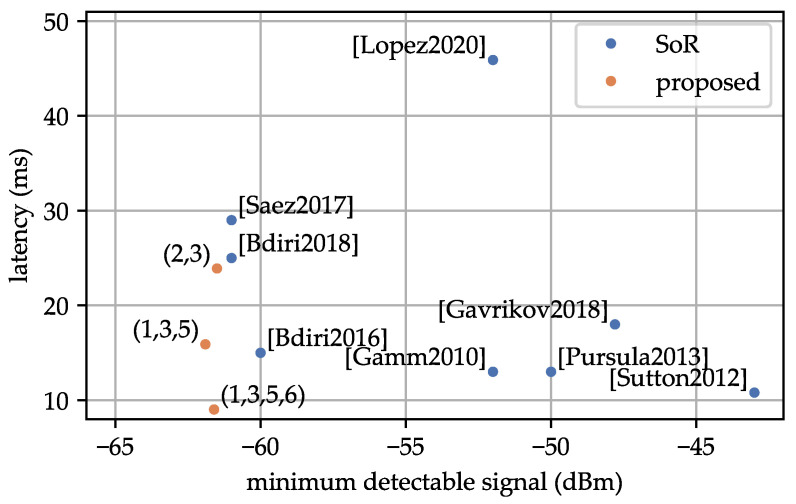
Comparison between MDS and latency of the state of research and the proposed WuRx design. References: [2] [Gamm2010], [10] [Bdiri2018], [14] [Saez2017], [15] [Bdiri2016], [18] [Lopez2020], [19] [Pursula2013], [20] [Gavrikov2018], [21] [Sutton2012].

**Table 1 sensors-23-08188-t001:** Summary of the wake-up receiver in the state of research. Our proposed wake-up receiver designs are shown in the first two columns. The table is sorted by the minimum detectable signal (MDS) column.

Ref.	(1,3,5)	(1,3,5,6)	[10]	[14]	[15]	[16]	[17]	[2]	[18]	[19]	[20]	[21]
MDS (dBm)	−61.9	−61.6	−61	−61	−60	−53.8	−53.2	−52	−52	−50	−47.8	−43 ^1^
Consumption ^2^ (µW)	5.71	5.71	7.2	70.2	7.5	9	8.34 ^3^	7.8 ^3^	10.8	20.4 ^3^	18.5	8.25
WuPt Duration (ms)	14.0	9.02	25	29 ^4^	15	—	—	13	45.9	13	18 ^4^	10.8 ^4^
RF (MHz)	868	868	868	838,868	868	868	433	868	2400	2450	2404,2474	433
Detector ^5^	VD	VD	VD	VD, d	VD	VD, d	VD	VD	VD	VD, b	VD, d	VD
Diode ^6^	7630	7630	285x	285x	285x	285x	285x	285x	—	286x	285x	285x
Amplification	no	no	OA	OA	OA	no	no	no	no	no	OA	no
LFPM ^7^	33	33	33	33	33	32	32	32	33	30	33	30
LF (kHz)	25.7	25.7	17	20	18	—	125	125	16	—	11.4	125
Data Rate (kbit/s)	3.21	3.21	4	1.2 ^4^	4.57	—	—	2.97	—	2.73	1.44	2.73
Addressing	16 bit	8 bit	16 bit ^8^	16 bit	16 bit	—	16 bit	16 bit	—	—	16 bit	8 bit ^8^
ON/OFF mode	50%	50%	11% ^9^	—	11% ^9^	—	—	100% ^9^	—	—	—	—

— No information provided. ^1^ Referring to peak power. ^2^ Only WuRx components. ^3^ Power consumption calculated from supply current and 3 V supply voltage. ^4^ Estimated from publication’s values and figures. ^5^ Single diode (SD), VD (voltage doubler), dual (d), biased (b). ^6^ SMS7630, HSMS-285x, HSMS-286x series. ^7^ AS3930, AS3932, AS3933. ^8^ Manchester coding. ^9^ Estimated from LFPM power consumption. Abbreviations: LF (low frequency), LFPM (low-frequency pattern matcher), MDS (minimum detectable signal), OA (operational amplifier), RF (radio frequency), WuPt (wake-up packet).

**Table 2 sensors-23-08188-t002:** State-of-the-art low-frequency profile of the AS3933.

Parameter	Value
**Wake-Up Packet**	
Carrier Burst	3.42 ms
Preamble	4.3 ms + 0.4 ms separation bit
Addressing	16 bit, Manchester
WuPt Duration	21.8 ms
**AS3933 Timing**	
Clock Source	RCO, 32.768 kHz
Band	5
LF	18.72 kHz
Data Rate	2.34 kbit/s
Modulation Ratio	8 carrier/bit
**Data Slicer**	
Fast Envelope	for 2.184 kbit/s
Slow Envelope	for 3.5 ms preamble
Mode	variable
Hysteresis	symmetrical, 40 mV
**AS3933 Power Saving**	
G_BOOST	enabled
ON/OFF mode	disabled

**Table 3 sensors-23-08188-t003:** Analysis of frequency bands and data rates.

Band	$f_{\mathrm{LF}}$ (kHz)	$f_{\mathrm{LF}}/f_{\mathrm{bit}}$	$t_{\mathrm{WuPt}}$ (ms)	MDS ($\mu V_{\mathrm{PP}}$)
Band 5	18.7	4	14.3	145
* Band 5	18.7	8	21.8	138
Band 4	29.1	8	16.5	157
Band 3	52.4	8	10.6	143
Band 3	52.4	16	15.6	139
Band 2	87.4	16	10.8	132
Band 1	131.1	16	8.30	164

* state-of-the-art profile.

**Table 4 sensors-23-08188-t004:** Measured MDS level in $\mu V_{\mathrm{PP}}$ for different data slicer configurations.

	Symmetric,	Positive,	Symmetric,	Positive,
	**40 mV**	**40 mV**	**20 mV**	**20 mV**
variable	* 138	245	134	223
absolute	181	219	192	212
reduced absolute	131	146	129	144

* state-of-the-art profile.

**Table 5 sensors-23-08188-t005:** Alteration of the RC oscillator frequency and measured MDS.

$f_{\mathrm{clk}}$ (kHz)	25.0	27.5	30.0	* 32.8	35.0	37.5	40.0	42.5	45.0
$f_{\mathrm{LF}}$ (kHz)	14.3	15.7	17.1	18.7	20.0	21.4	22.9	24.3	25.7
$f_{\mathrm{bit}}$ (kbit/s)	1.79	1.96	2.14	2.34	2.50	2.68	2.86	3.04	3.21
$t_{\mathrm{WuPt}}$ (ms)	29.9	27.2	24.9	22.8	21.3	19.9	18.7	17.6	16.6
**MDS** ($\mu V_{\mathrm{PP}}$)	156	148	141	138	141	134	133	135	133

* state-of-the-art profile.

**Table 6 sensors-23-08188-t006:** Summary of measurements with ON/OFF mode. Measured supply current and random packet loss for different carrier burst lengths $t_{\mathrm{CB}}$.

$t_{\mathrm{off}}$ (ms)	0	1	2	4	8
$I_{\sup}$ (μA)	3.45	2.60	2.31	2.08	1.92
**Random packet loss**					
$t_{\mathrm{CB}}=3.42\;\mathrm{ms}$	* 0%	0%	26%	57%	76%
$t_{\mathrm{CB}}=4.27\;\mathrm{ms}$	0%	0%	12%	44%	68%
$t_{\mathrm{CB}}=5.55\;\mathrm{ms}$	0%	0%	0%	33%	62%
$t_{\mathrm{CB}}=7.26\;\mathrm{ms}$	0%	0%	0%	16%	54%

* state-of-the-art profile.

**Table 7 sensors-23-08188-t007:** Measurements with different supply voltages.

$V_{\sup}$ (V)	$I_{\sup}$ (μA)	$P_{\sup}$ (μW)	MDS ($\mu V_{\mathrm{PP}}$)
3.3	3.45	11.4	135
* 3.0	3.30	9.90	138
2.8	3.06	8.57	151
2.6	2.99	7.78	151
2.4	2.94	7.06	154

* state-of-the-art profile.

**Table 8 sensors-23-08188-t008:** Measurement of the AS3933’s input impedance using the RSSI reading.

Band	* 5	1
$f_{\mathrm{LF}}$ (kHz)	18.7	131
Reference Level ($\mu V_{\mathrm{PP}}$)	271	183
$R_{s}$ (kΩ)	160	62
Level with $R_{s}$ ($\mu V_{\mathrm{PP}}$)	552	334
$R_{\mathrm{in}}$ (kΩ)	155	74.3

* state-of-the-art profile.

**Table 9 sensors-23-08188-t009:** Length and hexadecimal pattern of an example WuPt in band 5, with T_HBIT = 13 and Manchester coded address 0x5C2F.

Part	Duration (ms)	Pattern
Carrier burst	3.42	AAAAAAAAAAAAAAAAAAAAAAAAAAAAAAAA
Preamble	4.70	0000AAAA0000AAAA0000AAAA0000AAAA0000AAAA0000
Address (5)	3.42	0000AAAAAAAA00000000AAAAAAAA0000
Address (C)	3.42	AAAA0000AAAA00000000AAAA0000AAAA
Address (2)	3.42	0000AAAA0000AAAAAAAA00000000AAAA
Address (F)	3.42	AAAA0000AAAA0000AAAA0000AAAA0000

**Table 10 sensors-23-08188-t010:** Measurement results of multiple WuRxs. The LF MDS was estimated by the measured values.

PCB	$\gamma^{\mathrm{OC}}$ (mV/μW)	MDS (dBm)	LF MDS ($\mu V_{\mathrm{PP}}$)
1	35.8	−61.4	113
2	20.8	−59.0	116
3	29.8	−60.5	116
4	22.8	−58.8	132
5	20.7	−58.7	122

**Table 11 sensors-23-08188-t011:** Summary of the parameters of the SoAP and RF WuRx hardware.

Parameter	Value
**Performance**	
MDS	−62.0 dBm
Supply Voltage	3.0 V
Supply Current	3.30 μA
Power Consumption	9.90 μW
WuPt Duration	21.8 ms
**Envelope Detector**	
Matching	LC, 868 MHz
Type	voltage doubler
Diodes	SMS7630
Band-Pass Filter	$C_{\mathrm{LP}}=47\mathrm{pF},C_{\mathrm{HP}}=1\mathrm{nF}$
**Wake-Up Packet**	
Carrier burst	3.42 ms
Preamble	4.70 ms
Addressing	16 bit, Manchester
**AS3933 Timing**	
Clock Source	RCO, 32.768 kHz
Band	5
LF	18.72 kHz
Data Rate	2.34 kbit/s
Modulation Ratio	8 carrier/bit
**Data Slicer**	
Fast Envelope	for 2.184 kbit/s
Slow Envelope	for 3.5 ms preamble
Mode	variable
Hysteresis	symmetrical, 40 mV
**AS3933 Power Saving**	
G_BOOST	enabled
ON/OFF mode	disabled

**Table 12 sensors-23-08188-t012:** Proposed improvements and measured parameters of the different profiles (1)–(6).

Improvement	(1)	(2)	(3)	(4)	(5)	(6)
$t_{\mathrm{off}}$ (ms)	**1**	**2**	0	0	0	0
Supply Voltage (V)	3.0	3.0	**2.4**	3.0	3.0	3.0
Data Slicer	variable	variable	variable	**absolute**	variable	variable
$f_{\mathrm{RC}}$ (kHz)	32.8	32.8	32.8	32.8	**45**	32.8
Address Length (bit)	16	16	16	16	16	**8**
Power Consumption (μW)	7.38	6.48	7.32	9.90	10.2	9.90
MDS (dBm)	−61.7	−61.4	−61.8	−59.7	−62.1	−61.8
WuPt Duration (ms)	21.8	23.9	21.8	17.5	14.0	15.0

The modified parameters are highlighted in bold.

**Table 13 sensors-23-08188-t013:** Proposed combined improvements and measured parameters.

Improvements	(1,3)	(2,3)	(5,6)	(1,3,5)	(1,3,5,6)
$t_{\mathrm{off}}$ (ms)	**1**	**2**	0	**1**	**1**
Supply Voltage (V)	**2.4**	**2.4**	3.0	**2.4**	**2.4**
Data Slicer	variable	variable	variable	variable	variable
$f_{\mathrm{RC}}$ (kHz)	32.8	32.8	**45**	**45**	**45**
Address Length (bit)	16	16	**8**	16	**8**
Power Consumption (μW)	5.45	4.73	10.2	5.71	5.71
MDS (dBm)	−61.7	−61.5	−61.7	−61.9	−61.6
WuPt Duration (ms)	21.8	23.9	9.02	14.0	9.02

The modified parameters are highlighted in bold.

## Data Availability

The data presented in this study are openly available in FigShare at 10.6084/m9.figshare.20102303.

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
