# Peer review of "An Improved Wake-Up Receiver Based on the Optimization of Low-Frequency Pattern Matchers"

_sensors, 2023, doi:10.3390/s23198188_

Round 1

Reviewer 1 Report

The text has several parts, even subsections, that needs a full rewrite. The language is almost incomprehensible.

The text seem to be a compilation of different writers with different styles. A major editorial review is needed to make the text smooth.

Author Response

please see the appendix

Reviewer 2 Report

This paper on WuRx is a huge engineering work of great interest for the community. Even if this is not a huge fundamental novelty, this kind of paper really helps other researchers to understand the key parameters to optimize such systems.

This work is focusing on WuRx based on LFPM, which is restricting a bit the target, and a comment on that is that the positioning is not always perfectly clear: do the authors claim that their performance is better than any WuRx or LFPM-based WuRx ? Some other papers, like the one cited in reference [8] or others for instance from Kazdaridis et al, are presenting comparable or better performance, but with different approaches. Anyway, Fig 15 and 16 are a good way of referencing this work with regards to the SotA.
The discussion about MDS is interesting, ven though the conclusion is still not very clear to me. I understand the offset considering average power, but still results bellow -50 dBm with such Schottky diodes seem doubtfull to me and maybe difficult to reproduce. An analysis other a set of realizations should be preferable.

Some few typos to be fixed with a careful proofreading but globally the language is fine.

Author Response

please see the appendix

Reviewer 3 Report

This paper although it has potential and it show genuine research conducted by authors but its clear in many aspects. For example, sentences such as 336, 550 etc. Does not make sense  or very weak. Figures such as figure 10 in unclear you need to elaborate more. Figure 5 was adopted from anouther work ref. 6 not sure why authors used it. Some sentances  hazy,  weak and not clear.

The prposed work is too short and not clear. Can you please elaborate? 

My suggestion  to have a complete review  and editing try to explain exactly what you mean. Then have it reviewed  by a professional english editor.                                                                                                                   

Discussion part is too short and barly has a discussion.  Conclusion  can not be  part of a discussion  becaus ethis will confuse  the redear. Do you mean its a Conclusion  for the discussion  or for the whole paper ? 

  Refrences  too short and 3 of them are self cited which is okay but we need more reliable  Refrences. 

English editing by a professional  must be conducted 

Author Response

please see the appendix

Round 2

Reviewer 3 Report

I dont have any thing to add since my last review. Thanks

The english is okay